# Effect of Systemic Insecticides Applied via Drench on the Mortality of *Diaphorina citri* on Curry Leaf

**DOI:** 10.3390/insects14050422

**Published:** 2023-04-28

**Authors:** Julia Gabriela Aleixo Vieira, Emile Dayara Rabelo Santana, Leonardo Vinicius Thiesen, Thaís Fagundes Matioli, Pedro Takao Yamamoto

**Affiliations:** Department of Entomology and Acarology, “Luiz de Queiroz” College of Agriculture, University of São Paulo (ESALQ/USP), Piracicaba 13418-900, Brazil; emiledayara@usp.br (E.D.R.S.); leonardo.thiesen@usp.br (L.V.T.); thaisfagmatioli@gmail.com (T.F.M.); pedro.yamamoto@usp.br (P.T.Y.)

**Keywords:** neonicotinoids, chemical control, toxicity, trap plant, attract-and-kill system, thiamethoxam

## Abstract

**Simple Summary:**

Huanglongbing (HLB), a disease associated with the bacteria *Candidatus* Liberibacter spp. and transmitted by the Asian citrus psyllid (ACP) *Diaphorina citri* Kuwayama (Hemiptera: Psyllidae), is the primary and most destructive disease that affects citrus crops. The disease management tactics can target both the causal agent and the insect vector. Studies have demonstrated the potential of the curry leaf tree (*Bergera koenigii* L.) as a trap plant for *D. citri*, as it is more attractive than commercial citrus plants, in addition to being immune to the bacteria that cause HLB. The use of the trap plant technique, combined with systemic insecticides, creating an attract-and-kill system, can significantly increase the effectiveness of disease and pest management. The systemic insecticides thiamethoxam, thiamethoxam + chlorantraniliprole, and imidacloprid, frequently used in citriculture, were tested on adults of *D. citri* in the curry leaf tree by the drench method. The results demonstrated efficacy in controlling *D. citri*, showing the high mortality (>90%) of adults and a prolonged effect of biological activity after application in field and protected cultivation experiments. Low concentrations of insecticides affected *D. citri* oviposition. The results indicate that it is possible to combine the use of the curry leaf tree as a trap plant with systemic insecticides in citrus, increasing control efficacy and reducing the application of insecticides used in commercial groves.

**Abstract:**

Huanglongbing (HLB), the most serious disease in citriculture, is caused by the bacteria *Candidatus* Liberibacter spp., which is transmitted by the Asian citrus psyllid (ACP) *Diaphorina citri*. HLB is mainly controlled with insecticides, necessitating the development of alternative methods, e.g., the use of trap plants such as curry leaf *Bergera koenigii*, which is highly attractive to the ACP. We evaluated the effects of the main systemic insecticides used by citrus growers, applied via drench to adults of *D. citri* on the curry leaf tree. We tested the persistence of three pesticides: thiamethoxam, thiamethoxam + chlorantraniliprole, and imidacloprid in protected cultivation and the field condition at 7, 14, 28, 42, 56, 70, 98, and 154 days after the application. Different concentrations of insecticides containing the active ingredient thiamethoxam were tested on adults to determine the LC_10_ and LC_50_. Finally, we assessed the sublethal effects on the oviposition and development of *D. citri*. The insecticides controlled the adults for long periods. However, in the field experiment, from 42 days after application there was a decrease in mortality caused by pesticides applied via drench, while in the protected cultivation, mortality did not decline until the last day of evaluation. The median lethal concentration (LC_50_) for thiamethoxam was 0.031 g of active ingredient per plant, and for thiamethoxam in a mixture, the LC_50_ was 0.028 g a.i. per plant. In the experiment with sublethal doses, *D. citri* did not oviposit on the treated plants. Our findings suggest that the attract-and-kill system using the curry leaf tree and systemic insecticides is effective for the control of *D. citri* and contributes to the integrated management of HLB.

## 1. Introduction

The Asian citrus psyllid (ACP), *Diaphorina citri* Kuwayama (Hemiptera: Psyllidae), is the insect vector of the phloem-limited bacteria *Candidatus* Liberibacter spp., which is associated with the main and most destructive citrus disease, Huanglongbing (HLB), or greening [1,2]. In the Americas, Ca. L. asiaticus (CLas) is the most widespread bacterium in citrus crops [3].

At present, no methods are available to cure the plants or reduce the symptoms and damage caused by the disease. Therefore, HLB is managed by preventing the infection of citrus plants [3,4,5]. Currently, chemical control, through contact insecticides applied by foliar spraying or systemic insecticides applied on the soil and on the trunks of young, non-producing citrus trees, is the main tool for managing the insect vector [6]. Systemic application, which controls the ACP for a longer period and has less impact on beneficial insects, is the preferred control method for *D. citri* on young trees [7,8].

Although insecticides are effective, they have several environmental consequences, such as killing non-target organisms, including pollinators and natural enemies [9]. The indiscriminate use of pesticides can select for resistant populations, causing outbreaks of secondary pests and the resurgence of the target pest [10,11,12,13,14]. In addition, pesticides pose risks to human health, compromising food safety through contamination [15]. Therefore, the management of Huanglongbing requires an integrated approach aimed at controlling *D. citri* and the disease, since alternatives to insecticides are essential in reducing the impact of the disease on the citrus industry [16]. Thus, the trap plant tactic has shown excellent potential for reducing the insect vector and for managing the disease [13,17,18,19].

The strategy behind trap plants consists of attracting insects or other organisms to protect the target crops from attack by pests, preventing them from causing significant damage and concentrating them in a certain part of the field where they can be controlled with other methods and at low cost [20,21,22]. Host plants suitable for food and the development of *D. citri* can function as a trap to attract, intercept, or reduce the movement of these pests to the commercial crop [17]. The plant species chosen is usually planted on the border of the main crop, allowing chemical applications to be concentrated where pest insects occur in larger numbers [21]. The main advantages of using trap plants for pest control are the reduction of insecticide use, cost reduction, and the preservation of natural enemies, improving the quality and productivity of the harvest [20,22]. In addition, for insect vectors of phytopathogens, a trap crop on the border helps to reduce the primary dissemination of the disease by preventing the insects from entering the interior of the crop, since adult psyllids prefer to colonize plants at the edge of an orchard [23].

The management of *D. citri* and HLB by means of trap plants has been studied extensively [13,17,18,19]. Field studies have demonstrated that the use of *Murraya paniculata* L. (Rutaceae) as a trap plant can reduce the population density of *D. citri* [17]. However, there are reports that *M. paniculata* can be infected with CLas, even though the bacterial titer is lower than that found in commercial citrus species [24]. A congener, *Bergera koenigii* L. (Rutaceae), popularly known as curry, has shown potential for use as a trap crop in commercial citrus orchards [13,18]. This species is a suitable host for the development of *D. citri*, in addition to being more attractive than the citrus plant and immune to the disease [13,18]. The shoots produced continuously by *B. koenigii* can help to maintain high populations of the psyllid, serving as an important reservoir for these insects [18]. Recently, the first report of the successful introduction of a Bt protein (Cry1Ba1) in curry leaf described strong effects on the mortality of *D. citri* [16]. The use of Bt curry leaf may be an alternative tool for the management of HLB and the insect vector by citrus growers.

The main constraint on the use of trap crops is the inability to retain insects in the orchard [25]. Methods are needed to prevent the insects from dispersing to commercial orchards [25]. The application of suitable insecticides to these alternative hosts and to citrus plants may help to slow the spread of HLB by reducing the incidence of *D. citri* in the field [18]. Neonicotinoid insecticides are often applied to the soil around young citrus plants directly via drench. The insecticide is translocated through the xylem from the roots to the leaves [26], ensuring optimal coverage. Among the products used in drench application, the systemic insecticides thiamethoxam and imidacloprid are more effective, and are often used to reduce the population of *D. citri* populations in citrus plants [7]. The use of systemic insecticides allows the implementation of an attract-and-kill system, has less of an impact on natural enemies, and can be used in conjunction with biological-control agents when applied to the soil [7,8,27]. Hence, the use of *B. koenigii* as a trap plant in commercial citrus crops can help to reduce the incidence of the disease, and is even more effective when combined with other control methods such as transgenics [16], biological control with parasitoids [28], and systemic insecticides [6,8].

The use of *B. koenigii* as a trap plant in an attract-and-kill system depends on the possibility of eliminating the insect vector attracted to the plants. Therefore, it is necessary to know the lethal and sublethal effects of the main insecticides on *D. citri* used in citrus cultivation when applied to the trap plant. This study determined the lethal and sublethal effects and the persistence of systemic insecticides applied via drench to curry leaf trees as a potential alternative trap plant, aiming for the management of the insect vector of HLB.

## 2. Materials and Methods

### 2.1. Diaphorina citri Rearing Colony

The rearing colony of *D. citri* was maintained in climate-controlled rooms at 25 ± 2 °C, 60 ± 10% RH, and 14:10 h (L:D) at the Integrated Pest Management Laboratory of the Department of Entomology and Acarology, Luiz de Queiroz College of Agriculture (ESALQ/USP), Piracicaba, Brazil. Orange jasmine seedlings (*M. paniculata*) purchased commercially were used to rear the insects [29,30].

First, the plants were pruned to a height of approximately 25 cm, and when they produced shoots (2 to 3 cm long), they were placed in rearing cages (45 cm x 50 cm), serving as a substrate for feeding and oviposition by females of *D. citri* [28]. Approximately 300 adult specimens at the beginning of their reproductive period (12 days after hatching) were placed in each cage. The insects remained in the cages for seven days to allow time for oviposition. After the oviposition period, the adults were removed using a manual sucker, and the plants containing the eggs remained in the cages to support the insects’ development.

### 2.2. Insecticides

The commercial insecticides tested are registered for the control of *D. citri* in citrus crops in Brazil (Table 1).

### 2.3. Persistence of Systemic Insecticides Applied via Drench to Kill Diaphorina citri under Protected Cultivation and Field Conditions

Three systemic insecticides registered for citriculture were tested (Table 1). The insecticides were applied via drench, with a volume of 50 mL of spray per plant/pot, the same approach used by citrus growers [31]. For the protected cultivation test, the curry seedlings were planted in pots containing 6.5 to 7 kg of soil. After the insecticide application, the plants remained in a protected cultivation, and were irrigated three times a week, with about 150 mL of water per pot during the period of the experiment. For the field test, the plants were treated with the insecticides, and after seven days were planted in the experimental area.

In the test under protected conditions, an experimental block design was used, with randomized plots with six replications per treatment (insecticides). In the field test, the design was completely randomized, consisting of five replications per treatment. Ten insects up to seven days old per repetition (plant) were used, and were changed at each confinement in the curry leaf.

In the test on protected plants, *D. citri* adults were confined in voile-fabric cages on the plants at 7, 14, 28, 42, 56, 70, 98, and 154 days after application of the insecticides (DAA). In the field experiment, confinement started on the 14th DAA, due to the previous treatment with insecticides and the subsequent installation of curry seedlings in the area. After confinement, the mortality was evaluated on the third day.

### 2.4. Determination of Median Lethal Concentration (LC_50_) of Systemic Insecticides in Curry Leaf in Protected Cultivation

The curry leaf trees were treated with thiamethoxam + chlorantraniliprole and thiamethoxam, as these insecticides performed best in bioassay 1 (Section 2.3). Different concentrations that provided mortalities between 5 and 95% of the individuals were diluted in 50 mL of distilled water and applied via drench to each curry leaf. After seven days, ten insects were confined on each plant in cages made of voile. The mortality of *D. citri* was evaluated at 1, 2, 3, 5, and 7 days after confinement, in order to determine the best time to evaluate the effects of the insecticides. The design was completely randomized, in which each treatment (concentration) consisted of four replicates of 10 adult insects each. The concentrations used were determined using Finney’s equation [32], where the minimum and maximum concentrations were determined in preliminary tests.

### 2.5. Sublethal Effects of Systemic Insecticides on Diaphorina citri Oviposition

The curry leaf trees were transplanted into pots and pruned in order to obtain new leaves and shoots for the confinement of the psyllids. With the data from the protected cultivation bioassay, the LC_10_ values of thiamethoxam and thiamethoxam + chlorantraniliprole were used to evaluate the sublethal effects on *D. citri* and its development.

The insecticides were applied in 50 mL of distilled water. The LC_10_ values of the products were 0.002 + 0.001 g a.i. for thiamethoxam + chlorantraniliprole, and 0.0025 g a.i. for thiamethoxam. The plants were treated with the insecticides via drench (treatments); the control treatment consisted of distilled water only.

Twenty adults were confined on each curry leaf tree, using a voile cage that covered the entire length of the plant. After seven days of confinement, the insects exposed to the plants were removed, and the eggs deposited on each plant were counted. The design was in randomized blocks, consisting of three treatments (insecticides + control) and five replications with 20 insects each.

### 2.6. Statistical Analysis

A generalized linear model (GLM) with binomial distribution and probit link function (corrected for overdispersion when necessary) was developed using mortality data from each concentration. The model fit was verified using half-normal plot (‘hnp’) simulation graphics [32] and by the χ^2^ test. To estimate the lethal concentrations (LC) in protected cultivation, concentration-response curve assays were developed using 7-day mortality data and subjected to Probit analysis [33]. The LC_10_ and LC_50_ for each insecticide, as well as their 95% confidence intervals (CI 95%), were calculated using the ‘MASS’ package [34]. For field and protected cultivation assays, the persistence of the insecticide activity graphs were constructed using second-degree polynomial models. Persistence curves were compared using Tukey’s test (*p* < 0.05). To determine the sublethal effects of the insecticides, the data were subjected to a non-parametric Kruskal-Wallis’ test and compared using the Dunn’s test (95% probability). 

## 3. Results

### 3.1. Persistence of Systemic Insecticides Applied via Drench Affecting the Mortality of Diaphorina citri under Protected and Field Conditions

In the test of the persistence of the insecticides applied via drench on curry leaf tree under protected-cultivation conditions, high mortality was observed three days after the insects contacted the treated plants, and this continued for 154 days. There were no significant differences among the insecticides, only between the control treatment and the treatments containing insecticide (F = 105.57; df = 3; *p* < 0.001) (Figure 1). The mortality rate of insects confined on the treated plants remained constant throughout the evaluation period, with an average mortality above 80% in all evaluation periods (Figure 1).

The results of the field trial differed from those in the protected cultivation. As in the previous bioassay, there were differences between the plants containing the insecticides and the control treatment (F = 24.50, df = 3, *p* < 0.001). However, the control efficiency was above 80% up to 42 days after application, with a subsequent decline in insecticide activity over the 154 days of evaluation (Figure 2). No differences among the treatments were observed at 154 days after application (χ^2^ = 3.89; df = 3; *p* = 0.1976), indicating that these two insecticides applied via drench retained some residual activity. For imidacloprid, the persistence of biological activity in the field declined sharply after 70 days when applied using the drench method (Figure 2).

### 3.2. Determination of Median Lethal Concentration (LC_50_) of Systemic Insecticides in Curry Leaf Trees

For thiamethoxam and thiamethoxam + chlorantraniliprole, an evaluation time of three days showed the best distribution of mortality data in the concentrations tested, indicating good model adjustment parameters (Figure 3, Table 2 and Table 3).

The mortality data for thiamethoxam alone showed that the LC_10_ after three days was 0.007 g a.i. per plant, and the LC_50_ was 0.031 g a.i. per plant (χ^2^ = 2.39; df = 4; *p* > 0.05). For the mixture of thiamethoxam + chlorantraniliprole, the LC_10_ was 0.006 g a.i. per plant and the LC_50_ was 0.028 g a.i. per plant (χ^2^ = 7.85; df = 4; *p* > 0.05).

Thus, three days after the *D. citri* infestation of curry leaf contaminated with both insecticides, both the mixture and thiamethoxam alone showed similar LCs with overlapping confidence intervals, indicating no significant differences in the mortality of *D. citri*. These data also indicated that only the active ingredient thiamethoxam affected *D. citri* survival.

### 3.3. Sublethal Effect of Systemic Insecticides on Diaphorina citri Oviposition

No oviposition occurred in the treatments where the plants were treated with systemic insecticides, while in the control treatment the mean oviposition was 13.4 eggs per plant (Table 4). These data show that even at low concentrations (LC_10_), thiamethoxam and thiamethoxam + chlorantraniliprole affected the oviposition of *D. citri*.

## 4. Discussion

Studies to enable the use of chemical control in combination with alternative methods for the integrated management of *D. citri* and HLB are essential. In order to use the curry leaf tree as a trap plant, a system to eliminate the insect vector is needed, since curry leaf is highly attractive and supports the development of *D. citri* [18]. In view of the low impact of systemic insecticides applied via drench on non-target organisms [7,8,27], drench insecticide application and curry trap plants can be combined to more effectively reduce populations of the insect vector. Here, we showed that the systemic insecticides tested are highly effective in controlling ACP on the curry leaf tree, indicating that this is a viable method for managing HLB and reducing insecticide use in commercial citrus crops.

The persistence data for the insecticides collected in the protected cultivation and in the field showed differences in the residual period. This was probably a consequence of the different conditions of the two environments, since, in the field, the plants are exposed directly to wind, precipitation, and sunlight, among other factors. However, the insecticides tested in the field showed high persistence, controlling about 90% of adults up to 42 days after application, and gradually decreasing over time. Another study had similar results, in which thiamethoxam and imidacloprid applied via drench to citrus plants continued to kill more than 80% of *D. citri* adults for 50 to 60 days [8]. High ACP mortality (~80%) within 10 DAA was also observed with drench application of thiamethoxam and imidacloprid [35]. The mortality rate was maintained for 90 days in the semi-field and for 60 days in the field in citrus plants [35].

Here, the insecticides caused the high mortality of psyllids in the first confinement on curry leaf trees, although only 7 and 14 days after application. In studies with systemic insecticides, a gradual increase in insect mortality is usually observed; for instance, the high mortality (80%) of *D. citri* nymphs was reached only 20 to 25 days after application to *Citrus sinensis* L. Osbeck plants [8]. These differences can be explained by the use of different plant species, as it is possible that insecticides in the phloem or conductive tissue move differently in different species. Plants are living systems and are in a continuous state of physiological and biochemical flux, so considerable variations may occur in the cuticular penetration rate, metabolism, and phytotoxicity between plant species [36]. Abiotic conditions can influence the translocation of substances through the phloem, such as time of year, luminosity, temperature, humidity, and nutrition [37,38], as well as periods of severe drought [39]. Another study also found that insecticides applied to citrus plants caused high ACP mortality more slowly than when applied to curry leaf [40]. This difference may be related to the tendency of *D. citri* to feed more on the xylem in curry leaf, whereas in citrus plants it feeds on the phloem most of the time [40].

Thiamethoxam and thiamethoxam + chlorantraniliprole applied via drench showed a prolonged effect and better performance in controlling the ACP compared to imidacloprid. Although the residual period of imidacloprid is important for the control of *D. citri*, the data showed a gradual decrease in control efficiency. At 154 DAA, the ACP mortality in plants treated with imidacloprid (~25%) was close to that in the control treatment (~20%). This was caused by natural mortality in the field, while the other two insecticides containing the active ingredient thiamethoxam killed 50% of the adults. The direct application of systemic insecticides to the soil provides continuous and prolonged protection, and this method is widely used for young plants because *D. citri* preferentially colonizes young plants and shoots [41]. The high efficacy of the products tested has previously been reported for the control of *D. citri*, and thiamethoxam, regardless of the dosage (1.00, 1.25, and 1.50 g), was effective in controlling the ACP in curry leaf tree treated via drench and maintained in protected cultivation [40]. The mortality of the specimens was higher for thiamethoxam than for imidacloprid when evaluated after 7 days of each confinement; in addition, the highest dosage tested caused the highest mortality of ACP, decreasing after 56 DAA [40]. Thiamethoxam was more efficient against adults compared with the mortality caused by imidacloprid in tests at 4 and 6 weeks after treatment of citrus plants [42]. Similarly, in the present study, thiamethoxam performed better in controlling *D. citri* in curry leaf.

Our results demonstrated that low concentrations of the systemic insecticides generated high mortalities of *D. citri* on the young curry leaf tree. Another study found a mean lethal concentration of 0.0001 g L^−1^ for thiamethoxam applied in a liquid diet for insects in the laboratory [26]. Although the LC_50_ found by these authors is lower than that in the present study, the application via drench can be affected by other factors, such as heavy rainfall and irrigation systems that can cause soil leaching [43].

Sublethal effects on *D. citri* oviposition were observed at low concentrations (0.01 g mL^−1^). Similar results were found for imidacloprid (0.1 μg mL^−1^) applied by immersion in *Citrus aurantiifolia* (Christm.) cv. Swingle, which showed sublethal effects on reproduction, longevity, and the development of *D. citri* [44]. Imidacloprid has been proven to reduce fecundity, and the treatment of plants with sublethal concentrations has sharply reduced feeding, which drastically reduces the fitness of the insect [44,45]. It is common for adult insects with food restrictions, malnutrition, or low food availability to show delays and reductions in egg production [46]. Another study also observed that thiamethoxam and imidacloprid applied on the soil affected the feeding and behavior of *D. citri* on citrus plants, mainly interfering with the phloem ingestion phase and reducing the duration of tasting activity [6]. The two insecticides had a repellent effect on *D. citri*, causing adults to migrate from plants previously treated with insecticides [6]. These factors may be related to the failure of *D. citri* to oviposit on curry leaf tree treated with low concentrations of insecticides in our study.

In addition to the effectiveness of thiamethoxam, thiamethoxam + chlorantraniliprole, and imidacloprid in reducing the population of *D. citri*, these insecticides have the potential to reduce the inoculation of Ca. L. asiaticus. When applied to *C. sinensis* plants, the transmission of the bacteria was reduced by about 59% [31]. Thus, in addition to controlling the population of *D. citri*, insecticides can also be used to manage the disease, reducing the spread of HLB [19,31].

Studies that integrate the use of systemic insecticides and trap plants to control *D. citri* and the management of HLB are still few in number. The combination of *M. paniculata* as a trap plant with the application of thiamethoxam to control *D. citri* was tested previously [19]. The results showed a mortality rate of approximately 100% in the first seven days after application in newly established orchards, in addition to reducing the recapture rate of psyllids in yellow sticky traps and the number of psyllids found on citrus trees, reducing the incidence of HLB by 43% [19]. However, studies addressing the potential of curry as a trap crop for *D. citri* are preliminary and recent. The curry leaf tree is considered an alternative trap plant mainly because it demonstrates an immunity to the disease [13,47]. A laboratory study found that curry leaf is more attractive to the psyllid than citrus plants [18]. Field tests demonstrated the attractiveness and tendency of insects to remain on curry trap plants, in addition to finding larger numbers of nymphs and adults in non-commercial groves [48]. One factor that makes the curry leaf more attractive than citrus plants is the continuous flux of shoots, which may help to maintain *D. citri* populations at times when citrus plants do not have shoots [17].

The effects of systemic insecticides applied directly to the soil are still not fully known, and further studies are needed regarding the sublethal effects on natural enemies and non-target organisms in citrus crops. Studies indicate that endoparasitoids can be exposed to the active ingredient of neonicotinoid insecticides when they are in the development phase inside the host’s body [49]. The contact is through transovarian chemical residues, since the transfer of toxic substances is mediated by the host’s physiology and is initially mediated by penetration through the cuticle of the parasitoid or by the ingestion of food contaminated with pesticides [49]. Therefore, systemic insecticides can affect the parasitoid, requiring further studies with the natural enemies that control ACP populations in the psyllid in the field. The main biological-control agent for *D. citri* is the ectoparasitoid *Tamarixia radiata* (Waterston) (Hymenoptera: Eulophidae). Studies investigating the effects of systemic insecticides on parasitoids are only at their early stages. The next steps will be to conduct more detailed research and assess the feasibility of using biological control agents together with trap plant practices and the application of systemic insecticides via drench.

## 5. Conclusions

The systemic insecticides tested here proved effective in eliminating *D. citri* on curry leaf trees, enabling the use of the attract-and-kill system. The insecticides containing the active ingredient thiamethoxam provided control for a prolonged period. The insecticides showed high persistence in the field, obtaining a control rate of over 90% up to 42 DAA, then gradually decreasing. Low concentrations of thiamethoxam and thiamethoxam + chlorantraniliprole showed effects on the oviposition of *D. citri*.

These advances can contribute to the use of other control methods that are synergistic and compatible, such as the use of the biological agent *T. radiata* [28] and the application of systemic insecticides in trap plants as an attract-and-kill system for the attraction and elimination of insect vectors [19], contributing significantly to the integrated management of pests and diseases in citrus crops.

## Figures and Tables

**Figure 1 insects-14-00422-f001:**
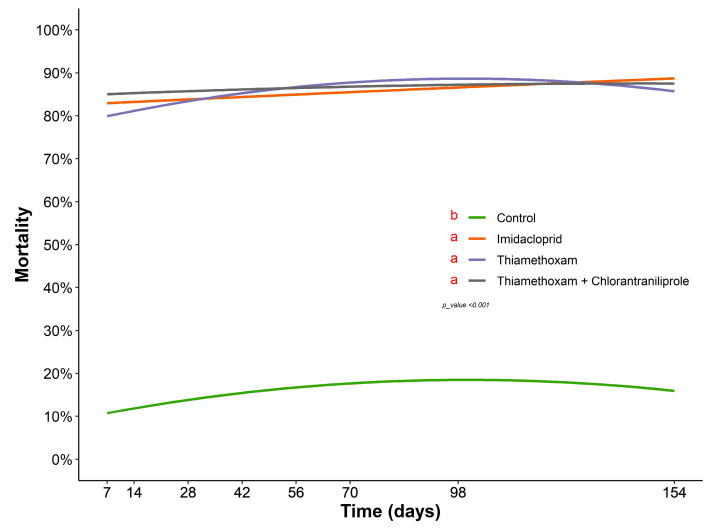
The persistence of insecticides applied via drench in curry leaf tree maintained in protected cultivation conditions for 154 days. The means of the treatments described in the legend, followed by different letters, differ significantly by Tukey’s test (*p* < 0.05).

**Figure 2 insects-14-00422-f002:**
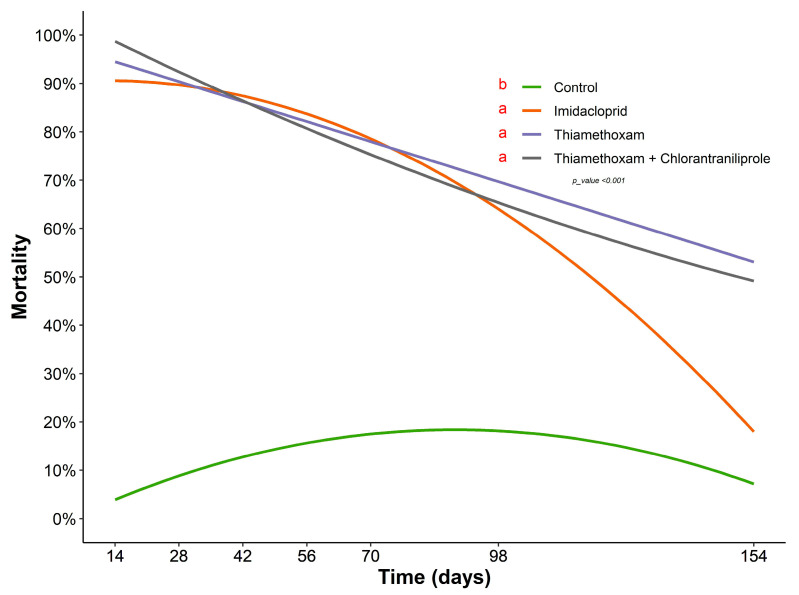
The persistence of insecticides applied via drench in curry leaf tree transplanted in the field over 154 days. The means of the treatments described in the legend, followed by different letters, differ significantly by Tukey’s test (*p* < 0.05).

**Figure 3 insects-14-00422-f003:**
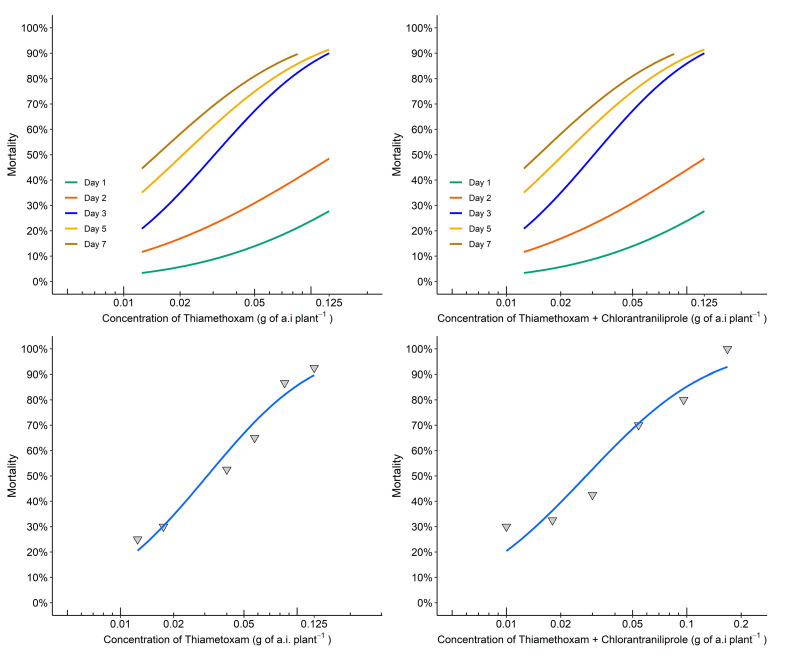
Concentration-response curve for the effects of the thiamethoxam and the thiamethoxam + chlorantraniliprole mixture on *Diaphorina citri* mortality over seven days of evaluation. Below, the isolated blue curves represent the distribution of mortality data evaluated on the third day, indicating a good fit of the model. The triangle point down means the observed mortality after three days.

**Table 1 insects-14-00422-t001:** Active ingredient, trade name, manufacturer company, chemical group, IRAC MoA, and recommended concentration of the main insecticides applied via drench in citrus crops to control *Diaphorina citri*.

Active Ingredient	Trade Name	Manufacturer Company	Chemical Group	IRAC MoA	Concentration (g a.i./Plant)
Thiamethoxam	Actara^®^ 25 WG	Syngenta^®^ São Paulo, SP–Brazil	Neonicotinoid	4A	0.25
Thiamethoxam + Chlorantraniliprole *	Durivo^®^ 30 SC	Syngenta^®^ São Paulo, SP-Brazil	Neonicotinoid + Diamide	4A + 28	0.2 + 0.1
Imidacloprid	Provado 20 SC	Bayer^®^ São Paulo, SP-Brazil	Neonicotinoid	4A	0.3

* Thiamethoxam + chlorantraniliprole = product in mixture.

**Table 2 insects-14-00422-t002:** Time (days), number of insects tested (N), and lethal concentration (LC, g a.i. plant^–1^) to kill 10 and 50% of *Diaphorina citri* adults contaminated with thiamethoxam on curry leaf trees.

Time (Days)	N	LC_10_ (CI_95%_) *	LC_50_ (CI_95%_) *	Slope (±SE) *	χ^2^ (df) *	*p*
1	320	0.035 (0.023–0.052)	0.375 (0.131–1.074)	1.24 (±0.32)	5.45 (5)	0.36
2	320	0.010 (0.005–0.021)	0.135 (0.076–0.239)	1.15 (±0.25)	8.94 (5)	0.11
3	240	0.007 (0.005–0.012)	0.031 (0.025–0.038)	2.09 (±0.28)	2.39 (4)	0.66
5	320	0.004 (0.002–0.008)	0.021 (0.016–0.027)	1.75 (±0.26)	7.54 (5)	0.18
7	250	0.003 (0.001–0.007)	0.015 (0.010–0.022)	1.68 (±0.36)	8.17 (4)	0.09

* CI: Confidence interval with 95% probability; SE: Standard error; df: Degrees of freedom.

**Table 3 insects-14-00422-t003:** Time (days), number of insects tested (N), and lethal concentration (LC, g a.i. plant^–1^) to kill 10 and 50% of *Diaphorina citri* adults contaminated with thiamethoxam + chlorantraniliprole on curry leaf trees.

Time (Days)	N	LC_10_ (CI_95%_) *	LC_50_ (CI_95%_) *	Slope (±SE) *	χ^2^ (df) *	*p*
1	280	0.005 (0.002–0.018)	0.197 (0.081–0.476)	0.81 (±0.21)	2.56 (4)	0.63
2	280	0.007 (0.004–0.012)	0.041 (0.032–0.052)	1.65 (±0.23)	8.63 (4)	0.07
3	270	0.006 (0.003–0.009)	0.028 (0.022–0.035)	1.87 (±0.24)	7.85 (4)	0.10
5	280	0.004 (0.002–0.008)	0.022 (0.017–0.029)	1.76 (±0.24)	5.91 (4)	0.21
7	280	0.003 (0.001–0.007)	0.015 (0.010–0.022)	1.67 (±0.24)	5.88 (4)	0.21

* CI: Confidence interval with 95% probability; SE: Standard error; df: Degrees of freedom.

**Table 4 insects-14-00422-t004:** Mean number of eggs laid by *Diaphorina citri* on curry leaf tree treated with LC_10_ levels of insecticides (g a.i. plant^–1^).

Insecticides	Concentration (a.i. Plant^–1^)	Oviposition (Mean ± SE) *
Control	-	13.4 ± 5.4 a
Thiamethoxam	0.0025 g	0 ± 0 b
Thiamethoxam + Chlorantraniliprole	0.0020 + 0.0010 g	0 ± 0 b

* Treatments followed by the same letter do not differ from each other by Dunn’s test (*p* < 0.05).

## Data Availability

The data presented in this study are available on request from the corresponding author.

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
