# Peer review of "Effect of Systemic Insecticides Applied via Drench on the Mortality of Diaphorina citri on Curry Leaf"

_insects, 2023, doi:10.3390/insects14050422_

Round 1

Reviewer 1 Report

A well-written and relevant manuscript on the use of trap plants combined with systemic insecticides for the management of a serious citriculture disease.

Figure 3 need some better explanation in the figure text. What does the lower curves show? Are they 72h?

Reviewer 2 Report

The paper describes a study conducted to develop an attract-and-kill strategy against Asian citrus psyllid by using an alternative host plant (curry leaf tree) treated with an insecticide. This is a very well-executed study and the manuscript is ready for publication. The Introduction section provides enough background information to set the stage for the rest of the paper. The methods are described in detail.

I do have a question about the Results section: it seems to me that the wrong graph is presented in Figure 2. The authors state that there was a decrease in mortality after day 42 in the open-field experiment, but I do not see it on the graph. Figures 1 and 2 are identical.

The Discussion section is very interesting, I appreciated how well the authors placed their findings in the context of the existing knowledge. Only one typo on line 262: more effectively does not require a hyphen. 

Excellent work, I think it should be accepted for publication after addressing these minor issues.
